# Intelligent Control Method of Hoisting Prefabricated Components Based on Internet-of-Things

**DOI:** 10.3390/s21030980

**Published:** 2021-02-02

**Authors:** Yuhong Zhao, Cunfa Cao, Zhansheng Liu, Enyi Mu

**Affiliations:** 1College of Architecture, Civil and Transportation Engineering, Beijing University of Technology, Beijing 100124, China; zhaoyuhong@bjut.edu.cn (Y.Z.); caocunfa@emails.bjut.edu.cn (C.C.); 2The Key Laboratory of Urban Security and Disaster Engineering of the Ministry of Education, Beijing University of Technology, Beijing 100124, China; 3Department of Land Economy, University of Cambridge, Cambridge CB2 1TN, UK; em781@cam.ac.uk

**Keywords:** IoT, LoRa, PC, hoisting, IMU

## Abstract

Prefabricated buildings are widely used because of their green environmental protection and high degree of industrialization. However, in construction process, there are some defects such as small wireless network coverage, high-energy consumption, inaccurate control, and backward blind hoisting methods in the hoisting process of prefabricated components (PC). Internet-of-Things (IoT) technology can be used to collect and transmit data to strengthen the management of construction sites. The purpose of this study was to establish an intelligent control method in the construction and hoisting process of PC by using IoT technology. Long Range Radio (LoRa) technology was used to conduct data terminal acquisition and wireless transmission in the construction site. The Inertial Measurement Unit (IMU), Global Positioning System (GPS), and other multi-sensor fusion was used to collect information during the hoisting process of PC, and multi-sensor information was fused by fusion location algorithm for location control. Finally, the feasibility of this method was verified by a project as a case. The results showed that the IoT technology can strengthen the management ability of PC in the hoisting process, and improve the visualization level of the hoisting process of PC. Analysis of the existing outdated PC hoisting management methods, LoRa, IMU, GPS and other sensors were used for data acquisition and transmission, the PC hoisting multi-level management and intelligent control.

## 1. Introduction

In recent years, prefabricated buildings developed rapidly with the advantages of industrial component production, convenient site installation, and green construction environment [1]. With the increase of prefabricated buildings, the safety problems of construction process are now more prominent. Prefabricated building construction site has the challenges of backward construction management and a low level of visualization [2,3]. Domestic and foreign scholars conducted extensive research on assembly construction management. Dave et al. [4] developed a communication framework based on the IoT, to strengthen lean construction management, and used tracking technology for Radio Frequency Identification (RFID), GPS, and other key components of the IoT, to track the whole process status of workers, materials, and equipment. Ko et al. [5] proposed a cost-effective material management and tracking system based on integrated RFID cloud computing services, which realized the omni-directional automatic tracking of materials. Xuetong Wang et al. [6] proposed a conceptual framework of an Intelligent Construction System for Prefabricated Buildings based on the IoT (ICSPB-IoT), and proved the feasibility of the realization of ICSPB-IoT through case studies. Zhiliang and his team, on the basis of overall building information management, also conducted research on prefabricated buildings, and put more emphasis on intelligence, on the basis of informatization [7]. He established an intelligent production planning and control system for residential parts and built a material management system based on mobile terminals for the construction site [8]. Zhao et al. [9] proposed a method based on the combination of cloud computing, Building Information Modeling (BIM), and IoT, to solve the problem of delayed information transmission on the construction site. Xu et al. [10] proposed a cloud-based IoT integration platform in view of the insufficient adoption of information technology in prefabricated buildings, which impedes the improvement of their efficiency. The integrated platform enables Small and Medium Enterprises (SMEs) to adopt IoT technology economically and flexibly. The above research aimed at the production of PC and construction site management, combined with RFID technology, BIM technology, cloud computing and GPS, and other information technologies; conducting a construction management framework of prefabricated buildings; tracking from production of PC to construction site and full cycle information management. However, there are few experimental applications of the latest technology of the IoT in prefabricated buildings. Moreover, the research on the automated construction management of the IoT is in the initial stage.

Meanwhile, with the rapid development of new technologies in the IoT, the research and application of LoRa and other Low Power Wide Area Network WAN (LPWAN) are gradually increasing. Knoll et al. [11] applied LoRa technology to directly monitor the air pollution value, which solved the problems of high monitoring cost and low precision at present, and conducted the test in downtown Graz. Mohammed et al. [12] developed and tested the service model of “intelligent street lamp” based on LoRa. Catherwood et al. [13] proposed a biomedical stripe diagnosis system where the adoption of LoRa based e-reader could provide the next generation of community-based intelligent diagnosis and disease management. Polonelli et al. [14] in Italy designed LoRaWAN-based wireless components for human structural health monitoring to measure and track cracks in concrete and other building materials. Shinan et al. [15] applied the Narrow Band Internet of Things (NB-IoT) technology in prefabricated building construction, created specific application plans, strengthened the management of PC on the construction site, and provided a strong reference for the application of the IoT technology in the construction industry. LPWAN, represented by the LoRa technology, was studied and applied in environmental monitoring, medical treatment, intelligent street lighting, building monitoring, and other fields [16,17,18]. However, there is a lack of relevant research on the application of LPWAN in the field of intelligent management of building construction and building automation. Moreover, there is a lack of research on the intelligent control of PC hoisting, based on LPWAN. Aiming at the lack of intelligent control of PC hoisting, this study used LoRa technology to integrate multiple sensors to strengthen intelligent control of PC hoisting. This study analyzed the application of information technology in the field of architecture, studied the fusion method of LPWAN and multi-sensor information integration, introduced the key role of IoT technology in realizing the intelligent control of PC hoisting, and analyzed the realization effect of multi-information fusion processing.

The remainder of the paper is organized as follows. Section 2 presents a literature review about previous work in related areas. Section 3 describes in detail the proposed multi-sensor sensing hoisting control method. Section 4 presents a sensor test and a field test of the proposed method applied to a real project. Section 5 discusses the results and concludes the paper.

## 2. Literature Review

### 2.1. IoT Analysis

IoT technology has developed rapidly, such as LoRa, NB-IoT, Ultra Wide Band (UWB), RFID, and other technologies, strengthening the connection between things and the Internet and between things [19,20]. IoT technology includes the intelligent application of transmission technology, sensors, actuators, and other internet-enabled devices in various fields [21]. The development of IoT technology provides a new opportunity for the upgrading and transformation of the construction industry [22]. At present, WiFi, Bluetooth, Zigbee, and other IoT technologies are widely used. These are characterized by convenient configuration, high security, and stability. However, the energy consumption and cost of WiFi technology are high, and the communication distance is short, so that WiFi technology is suitable for transmitting large files. The communication distance of Bluetooth technology is only about 10 m, so it cannot meet the needs of the extensive construction site. Zigbee’s biggest characteristic is ad-hoc network, with tens of thousands of network nodes. But Zigbee has the disadvantage of having very short transmission distances [15,23]. Therefore, the above IoT technologies are not suitable to be used in the large-scale and multi-component prefabricated building construction.

Compared to the above traditional IoT technology, LPWAN transmission distance is longer, energy consumption is lower, and is more suitable for prefabricated building construction. LoRa technology and NB-IoT technology are the two technologies that attract the most public attention in LPWAN. Comparing LoRa technology with NB-IoT technology, the conclusion that LoRa technology is more suitable for prefabricated building construction sites is drawn on the basis of the following advantages. (1) The network is flexible, and the construction unit can deploy the network by itself, without relying on operators. (2) The LoRa network has the advantages of lower cost, better transmission performance and longer transmission distance. (3) The power consumption of LoRa technology is lower in practical application [24,25]. Moreover, the information acquisition of PC in the construction site is characterized by multiple terminal nodes and small amount of data. In addition, extensive coverage, low energy consumption, and low-cost network technologies are needed, as information changes with the construction process. Therefore, LoRa technology has more applied advantages in the construction site of prefabricated buildings. LoRa technology is an information transmission technology based on LoRaWAN. Its essence is a spread spectrum modulation technology, combined with forward error-correcting coding, and digital signal processing technology. The LoRa technology adopts star network architecture with strong scalability, and has the lowest latency and lower network maintenance cost, as compared to the common network architecture [26,27]. Traditional LoRa technology is mainly applied in environmental monitoring, medical treatment, intelligent street lighting, and building monitoring. In recent years, the IoT technology is under constant innovation and many new breakthroughs were achieved.

### 2.2. IMU Application Analysis

IMU is widely applied in many fields such as automatic driving and robotics. IMU is generally a unit composed of three accelerometers, three gyroscopes, and magnetometers. Acceleration sensors can measure acceleration in three directions, and triaxial angular velocity can be measured by gyroscopes [28]. The IMU unit can be used to collect and monitor attitude data, and the object can be located by integrating twice with acceleration. However, due to the accumulated error of integration, the IMU has serious drift problem in long-distance positioning [29], which is caused by the accumulated drift of accelerometer error and gyroscope error in the long distance. After two times of integrations, the displacement would drift and the positioning would be inaccurate [30,31]. Aiming to solve this problem, Jimenez et al. [32] proposed a zero-velocity correction algorithm, which realized a multi-condition attitude detection algorithm, using information sources (accelerometer and gyroscope) and a low-pass filter. Meanwhile, aiming to solve the problem of long-distance position migration in IMU active positioning, IMU is mainly integrated with GPS, Beidou, GNSS, and other positioning methods for precise positioning of moving objects [33,34,35,36]. Chen proposed a fusion algorithm based on the Kalman filter for accurate positioning by integrating IMU and GPS algorithm for pedestrian positioning tracking [30]. This paper explored the integration of IMU and multi-sensors and the real-time upload and feedback of on-site information through the LoRa network, to strengthen the intelligent management of assembly building construction and hoisting.

### 2.3. Research Emphasis and Novelty

Although there are many applications of IoT in the field of prefabricated building construction, few studies were performed on the application of LoRa technology in PC construction management. Moreover, because of the backwardness of the communication in the operation of the PC, the communication on the swing and position of the PC in the process of hoisting is still in the form of megaphone and walkie-talkie [37]. Additionally, the performance of data acquisition and transmission on the construction site is backward, because more advanced transmission technology is not used in the field information acquisition and transmission [38], thus, there is no reliable information collection and transmission method in the hoisting process of PC, and the visual control of PC in the construction process is insufficient. Reliable information collection and transmission methods are lacking in the field of PC hoisting, which causes the obvious lack of visual control of PC during construction.

Through the above analysis, LoRa, IMU, and other multi-sensor fusion is used to establish the intelligent PC hoisting control method in this study. The method was employed to conduct meaningful investigations as follows. (1) Hoisting process data collected according to the IoT, and realizing real-time transmission of different types of data. (2) Based on the analysis of the comprehensive hoisting data, a hoisting construction management method of PC was proposed to guide the site construction personnel to conduct the hoisting construction. (3) The fusion positioning algorithm of IMU and GPS based on the Kalman filter was verified in the feasibility of hoisting positioning. (4) The site management and control of the PC hoisting was realized to help the hoisting operators and managers to quickly understand the status of the PC during hoisting. This innovative method, which integrated technologies like LoRa, IMU, GPS, integrated positioning, and others, improved the management level of PC hoisting, and is of great significance for future research on prefabricated building construction.

## 3. Proposed Method

### 3.1. LoRaWAN Architecture for Hoisting

LoRa technology networking is simple and flexible. The LoRaWAN network architecture includes four parts—LoRa chip, LoRa gateway, cloud server, service platform, and software; as is shown in Figure 1. 

The LoRa chip is used to collect real-time information about the hoisting of PC, including IMU, GPS, and barometers. LoRa gateway is used to receive the information transmitted from LoRa chip and wirelessly transmit it to the cloud server. The server analyzes and processes the information, then transfers it to the application service platform. The application platform is the final aggregation point of data, presenting data according to user requirements.

According to the LoRaWAN network architecture, the application scheme architecture framework of the LoRa technology in PC hoisting control is established. As is shown in Figure 2, IMU, GPS, barometer, RFID, and other sensors are used to collect data in the construction and hoisting process. Real-time upload of data through the LoRa terminal and the LoRa gateway. The construction site only needs to set up the LoRa gateway at a predetermined location, according to the construction site requirements, and the LoRa gateway can transmit the information collected from the LoRa module to the cloud server. The information uploaded to the cloud server is processed in the application layer, mainly including component information, attitude information, location information, etc.

Aiming to solve the problems existing in the hoisting of PC, in this paper, the application of IoT technology to strengthen the control of prefabricated parts hoisting is proposed. As is shown in Figure 3, the attitude information, including acceleration, velocity, and Euler Angle in the hoisting process of PC is collected through IMU attitude control unit. At the same time, the information collected by IMU is combined with GPS and barometer for fusion positioning to accurately monitor the real-time position of the PC in the hoisting process. The data collected by the above sensors are uploaded to the information layer through the LoRa technology, and real-time information transmission on the construction site is conducted through the LoRa technology.

### 3.2. IMU Information Collection

Acceleration sensors can measure acceleration in three directions. Gyroscopes can measure angular velocity in three axes. When IMU collects data, it generally first outputs the AD value, then converts the AD value to quaternion through transformation algorithm, and then calculates quaternion to Euler Angle through the transformation algorithm. The above introduces attitude control algorithm. However, the existing attitude sensing module, such as MPU6050, has a built-in program that can directly output quaternions instead of AD values. Moreover, a magnetometer is attached to the IIC interface to directly output an attitude packet, and get the attitude of the object under tests. The IMU unit is used to monitor the swinging attitude of components in the hoisting process in real time so as to distinguish the risks of hoisting. The IMU element consists of a built-in axis (coordinate) and a relative coordinate system whose vertical direction is the positive Z axis. By measuring the angle of the IMU data object in the hoisting process, the deflection angle in the hoisting process is calculated. By monitoring the attitude of components in the hoisting process, operators can judge whether the hoisting process would be dangerous due to the actual situation during the hoisting process, so as to terminate the hoisting process before something dangerous occurs.

### 3.3. Sensors Date Fusion Processing

Through IMU and GPS integrated positioning, the positioning accuracy is higher than that of the single GPS, and it is more suitable for building construction, which can better reduce the influence of GPS errors caused by field environmental factors. According to the characteristics of the construction site, based on the original IMU and GPS fusion algorithm [30], this study optimizes the design for the construction hoisting situation, and expands its application scenarios and methods. As the fusion algorithm due to the GPS height data is not accurate in the vertical direction (the Z axis) correction, fusion might not apply to the hoisting of the improved construction of the algorithm and the introduced barometer monitoring vertical height data. In this study, the algorithm is applied to make vertical location data more accurate, in order to meet the needs of the construction hoisting position.

#### 3.3.1. IMU Positioning Algorithm

The experimenters took due east as the X-axis and due north as the Y-axis, and took the vertical ground upward as the z-axis of the absolute coordinate system, to establish the absolute coordinate system W, according to the right-hand rule. The built in coordinate system of IMU is a relative coordinate system R, and the data of the relative coordinate system is converted into an absolute coordinate system through a transformation matrix. The transformation matrix is expressed as a unit quaternion:$Q=(q_{w},q_{1},q_{2},q_{3})$. Using state space representation, the system’s state vector can be given as
$$X_{k}={\left(P_{k},V_{k},O_{k}\right)}^{T}$$ where $P_{k}=(x_{k},y_{k},z_{k})$ is the displacement vector at time K, $V_{k}=({x^{\prime }}_{k},{y^{\prime }}_{k},{z^{\prime }}_{k})$ is the velocity vector at time K, and $O_{k}=(\varphi_{k},\theta_{k},\psi_{k})$ is expressed as the Euler angle at time K. $a_{k}^{w}$ is defined as the absolute acceleration at time K, and $a_{k}^{r}$ is the relative acceleration at time K, then:(1)$$a_{k}^{w}=Q_{k}\times a_{k}^{r}\times {Q^{\prime }}_{k}-G$$
where $Q_{k}=(q_{w,k},q_{1,k},q_{2,k},q_{3,k})$ is the rotation transformation matrix between the relative coordinate system R and the absolute coordinate system W is represented by quaternions. G=(0,0,g) is expressed as the gravitational acceleration vector.

Quaternions are converted to Euler angles by the conversion function Q2Euler(Q):(2)$$O_{k}=Q2Euler(Q)=\left(\begin{array}{c} atan2(2(q_{w}q_{1}+q_{2}q_{3}),1-2(q_{1}^{2}+q_{2}^{2}))\\ arcsin(2(q_{w}q_{2}-q_{3}q_{1}))\\ atan2(2(q_{w}q_{3}+q_{1}q_{2}),1-2(q_{2}^{2}+q_{3}^{2})) \end{array}\right)$$
(3)$$P_{k}=P_{\left(k-1\right)}+I\times T_{s}\times V_{\left(k-1\right)}+0.5T_{s}^{2}\times I\times a_{\left(k-1\right)}^{w}$$
(4)$$V_{k}=V_{\left(k-1\right)}+T_{s}\times I\times a_{\left(k-1\right)}^{w}$$
(5)$${\left(P_{k},V_{k}\right)}^{T}=\left(\begin{array}{cc} I & I\times T_{s}\\ 0 & I \end{array}\right){\left(P_{\left(k-1\right)},V_{\left(k-1\right)}\right)}^{T}+\left(\begin{array}{c} \frac{T_{s}^{2}}{2}\times I\\ T_{s}\times I \end{array}\right)a_{\left(k-1\right)}^{w}$$

In the equation, $T_{s}$ represents the sampling time interval of the sensor, and I represents the identity matrix.

#### 3.3.2. Zero-Velocity Stage Judgment 

Condition 1:(6)$$\left|a_{k}\right|={\sqrt{a_{k}^{r}{\left(1\right)}^{2}+a_{k}^{r}{\left(2\right)}^{2}+a_{k}^{r}{\left(3\right)}^{2}}}^{}$$
(7)$$C_{1}=\left\{\begin{array}{c} 1,thrhd_{a_{\min}}<\left|a_{k}\right|<thrhd_{a_{\max}}\\ 0,\;\;\;others \end{array}\right\}$$

In the equation, $thrhd_{a_{\min}}$ and $thrhd_{a_{\max}}$, respectively, represent the minimum threshold and the maximum threshold.

Condition 2:(8)$$\sigma_{a_{k}^{r}}^{2}=\frac{1}{2s+1}\sum_{j=k-s}^{k+s}\left(a_{k}^{r}-{\bar{a_{k}^{r}}}^{2}\right)$$
(9)$$C_{2}=\left\{\begin{array}{c} 1,\sigma_{a_{k}^{r}}^{2}<thrhd_{a_{\max}}\\ 0,\;\;others \end{array}\right\}$$

Condition 3:(10)$$\left|\omega_{k}\right|=\sqrt{\omega_{k}^{r}{\left(1\right)}^{2}+\omega_{k}^{r}{\left(2\right)}^{2}+\omega_{k}^{r}{\left(3\right)}^{2}}$$
(11)$$C_{3}=\left\{\begin{array}{c} 1,\left|\omega_{k}\right|<thrhd_{\omega_{\max}}\\ 0,\;\;others \end{array}\right\}$$

Through the above three conditions, the logical value of zero speed detection $C_{stop}$ = $C_{1}$&$C_{2}$&$C_{3}$, where $C_{1}$, $C_{2}$, $C_{3}$ are represented in Condition 1, 2, 3.

#### 3.3.3. Fusion Positioning

When considering the logical value of zero velocity test, the state matrix is expressed as:(12)$$\hat{X_{k}}={\left(\hat{P_{k}},\hat{V_{k}},\hat{O_{k}}\right)}^{T}=A\times X_{k-1}+C\times a_{k}^{\omega }+O_{k}$$
(13)$$A=\left(\begin{array}{ccc} I & I\times T_{s} & 0\\ 0 & I\times C_{s} & 0\\ 0 & 0 & 0 \end{array}\right)$$
(14)$$C={\left(\frac{T_{s}^{2}}{2}\times I,T_{s}\times I,0\right)}^{T}$$
(15)$$Q_{k}={\left(0,0,Q2Euler(Q_{k})\right)}^{T}$$

GPS height and speed data are not accurate, longitude and latitude is used as the actual measurement, and the latitude and longitude are converted into global coordinates using the conversion equation given by Dupree:(16)$$X_{gps}=\cos(\varphi )\sqrt{\frac{1}{{\left(\frac{\sin(\varphi )}{a}\right)}^{2}+{\left(\frac{cos(\varphi )}{c}\right)}^{2}}}\left[(lon1-lon0)\times \frac{\pi }{180}\right]$$
(17)$$y_{gps}=\sqrt{\frac{1}{{\left(\frac{\sin(\varphi )}{a}\right)}^{2}+{\left(\frac{\cos(\varphi )}{c}\right)}^{2}}}\left[(lat1-lat0)\times \frac{\pi }{180}\right]$$
(18)$$\varphi =2\pi -0.5(lat1-lat0)\frac{\pi }{180}$$

In the equation, (lon0, lat0) is the original point of geographical coordinates, (lon1, lat1) is the geographical coordinates of a specific point. $\left(x_{gps},y_{gps}\right)$ measured by GPS can be obtained through the above equation, and combined with the barometer $\left(z_{b}\right)$, then $D_{k}={\left(x_{gps},y_{gps},z_{b}\right)}^{T}$, defines the measurement matrix as:(19)$$H_{k}=\left(\begin{array}{ccccccccc} 1 & 0 & 0 & 0 & 0 & 0 & 0 & 0 & 0\\ 0 & 1 & 0 & 0 & 0 & 0 & 0 & 0 & 0\\ 0 & 0 & 0 & 0 & 0 & 0 & 0 & 0 & 1 \end{array}\right)$$

Next, Kalman filtering is performed. Calculated state prediction:(20)$$X_{k}={\hat{X}}_{k}+K_{k}(D_{k}-H{\hat{X}}_{k})$$
(21)$$K_{k}=\hat{\sum {}_{k}}H^{T}{\left(H\hat{\sum {}_{k}}H^{T}+R_{k}\right)}^{-1}$$
(22)$$\hat{\sum {}_{k}}=A\sum {}_{k-1}A^{T}+N_{k}$$
(23)$$\sum {}_{k}=(AI-K_{k}H)\hat{\sum {}_{k}}{\left(I-K_{k}H\right)}^{T}+R_{k}$$

The error of quaternion would cause the drift error of acceleration in the global frame. For this problem, the quaternion is modified to:(24)$$Q_{k}=\Delta Q_{k-1}\times Q_{k}$$
(25)$$\Delta Q_{k-1}=Euler2Q(0,0,K_{k}\times (D_{k}-H_{k}{\hat{X}}_{k}))$$

The measurement model is:(26)$$Z_{k}=H_{k}{\hat{X}}_{k}+n_{k}$$

$Z_{k}$ represents the predictive measurement, $n_{k}$ represents environmental noise, and the covariance matrix is $R_{k}=E(n_{k},n_{k}^{T})$.

When one of GPS with a barometer is unavailable, then ${\hat{x}}_{k}$ using the Kalman filter, directly to the next iteration calculation. The algorithm logic is shown in Figure 4.

## 4. Case Study of the Proposed Method

### 4.1. Case Background and Scenario Simulation

In this study, the example of the assembled floor construction and hoisting was used to analyze the practical application of the assembled floor construction and hoisting. The construction area of the second stage of Jintang Yuan was 71,172.43 square meters, including buildings 10~18, and its prefabricated structure was concrete shear wall structure. The PC of the project included prefabricated stairs, prefabricated wall panels, and prefabricated laminated floors. The total number of prefabricated members was 6174, including 656 prefabricated wall panels, 5224 prefabricated composite floors, and 294 prefabricated stairs. The number of PC in each building is shown in Table 1. In the table, Building No. 15 and Building No. 18 are divided into two parts due to their different floors, as is shown in Figure 5.

### 4.2. Hardware Selection and Development 

In this study, the power consumption of the IoT terminal is optimized at the level of hardware and algorithm system, and the optimization ideas mainly include three levels. The first level is the use of a more efficient DC–DC power management circuit. The second level is from the algorithm and embedded system program to maximize the sleep time, reduce the system empty running time, i.e., the use of long sleep, timing, and short time wake up mode, to minimize the power consumption of the system. The third level is the selection and update of the RFIC chipset, replacing the original SX1278 chipset with the lower power consumption SX1268 chipset, so as to further reduce the power consumption of the system.

The hardware selection is shown in Table 2, including the LoRa chip, MCU, etc. The IoT positioning terminal with improved DC–DC power supply and embedded system and its architecture analysis are shown in Figure 6. Meanwhile, the positioning module is added on the basis of the system. The system uses a modular addition approach, using a plug, where the positioning module is used when needed, to improve the positioning accuracy of the entire system. In the low power mode, the positioning module is not started, and the overall power consumption is reduced to the ideal state.

LoRa Gateway, located at the highest point of a tower crane on the construction site, collects perceptual data. The data collected from the sensor is transmitted to the gateway through LoRaWAN. The LoRa gateway can transmit the data to cloud servers and transfer the LoRaWAN protocol to TCP/IP protocol. LoRaWAN protocol to the TCP/IP protocol. In China, LoRa operates in the 470/510 M, which allows long-range coverage, with a bit rate ranging from 0.37 and 46.9 kbps [38]. The important parameters of the LoRa communication are shown in Table 3 and the prototype of the LoRa gateway used is shown in Figure 7.

The leysight power analyzer was used to conduct a complete power test and analysis of the improved positioning terminal; the instrumental analysis interface is shown in Figure 8. The power supply voltage was 3.3 V, and the current of on-chip temperature and humidity sensor was 39.7 mA under working conditions. In the case of single-point pulse signal transmission, its power consumption was 79.8 mA. The power consumption in the sleep mode was 53.7 uA. The data transmission efficiency of th eLoRa network is shown in Figure 9.

### 4.3. Analysis of the Hoisting Process

RFID tags are attached to the PC, which contain the corresponding PC number and other information for information verification. As is shown in Figure 10, the hoisting process of PC mainly included the following aspects of the gateway layout, device installation, component stacking, equipment sleep, determining the hoisting sequence, equipment activation, monitoring the hoisting process, information uploading, and module recovery. Through intelligent management of multiple sensor information collected in the hoisting construction process, the intelligent management of information checking, construction simulation, hoisting process monitoring, information uploading, and other aspects can be realized. The corresponding LoRa gateway were arranged in combination with the stacking of materials in the construction site, based on the LoRaWAN network architecture, and the LoRa information transmission network established in the construction site.

After the construction personnel confirms the hoisting, the active control module of the handheld device enters into the working state for real-time information transmission. Researchers select the appropriate spreader according to the prefabricated floor information, check whether the embedded hoisting ring on the assembly is firm, and lift it directly from the flat car after confirmation.

According to the attitude information collected by the active module, the tower crane operator monitors the deflection, tilt angle, and sling deflection angle of the PC, during the hoisting process.

Through the information collected by the integrated module, the operator of the tower crane checks the real-time location information of the PC on the equipment.

The path of PC in the hoisting process is composed of vertical movement, horizontal radial movement, and horizontal tangential movement. There are horizontal radial movement and horizontal tangential movement in the vertical hoisting of PC. The position of the PC is adjusted so that the PC is approximately directly above the installation point through the operator’s luffing and turning operation during the hoisting process, and then the vertical landing is conducted. It can be obtained through multi-sensor information fusion and algorithm processing, as shown in Figure 11. The path of the PC presents advanced path motion in the projection of the X-Y plane, and then tangential motion was conducted, and the corresponding tangential motion was adjusted in the process, and the deflection was conducted to the inside of the tangential motion. From the projection of the X-Z plane and the Y-Z plane, it could be seen that the vertical movement of the PC was accompanied by alternating tangential movement and radial movement, gradually shifting to the installation position. Meanwhile, it could be seen from the three-dimensional trajectory diagram of X-Y-Z that the PC presents a complex trajectory with three kinds of motions, continuously combined and alternately adjusted in the traveling path. The results presented in the figure are basically consistent with the pre-hoisting planning and the usual hoisting trajectory, which reflects the timely visual display and intelligent control effect of the multi-sensor fusion processing and transmission method, on the response of the PC in the hoisting process.

In the process of installation, the flatness of PC is corrected and the tilt angle information of PC is observed to avoid human error such as plumb line measurement. After the installation of PC is completed, field workers confirm through handheld devices and upload to the cloud server database for preservation, so as to facilitate subsequent review and acceptance. After uploading the information, workers reset the module information for subsequent hoisting.

## 5. Discussion and Conclusions

The rapid development of new generation technologies such as the IoT and artificial intelligence led the construction industry to upgrade to digital and intelligent construction.

IoT technology is the key technology to realize the transformation and upgrading of prefabricated building construction, as well as an efficient method for the management and control of PC hoisting system. This study analyses the construction hoisting status and the application of information technology, investigates the fusion method of LPWAN and multi-sensor information integration, introduces the key role of IoT technology in realizing intelligent control of PC hoisting, and analyses the realization effect of multi-information fusion processing. This method has the following advantages:

(1) Aiming for the application of IoT technology in the construction stage, the integration mechanism of LPWAN and multi-sensor was explored. The real-time data feedback provided a new idea for the realization of “real-time perception, intelligent analysis and intelligent decision” in PC construction.

(2) To apply the LoRa technology and multi-sensor fusion mechanism to hoisting of PC, we first performed the deep fusion of LoRa technology and multi-sensor to realize real-time perception and upload information, which lay a foundation for the deep visualization, real-time feedback, intelligent control, and accurate execution of prefabricated building construction. Meanwhile, the ability of depth information fusion to improve the optimization control of PC during the hoisting process is discussed.

(3) Improving the intelligentization of PC hoisting construction is a system engineering of multidisciplinary integration. This study took the intelligent control of PC hoisting based on the multi-information fusion transmission of the IoT as the core, realizing the multi-information fusion transmission and processing mechanism in the hoisting process, and continuously improving the ability of intelligent control of PC hoisting.

(4) Aiming at the problem of intelligent hoisting control of PC, an integrated method driven by intelligent sensing technology and intelligent algorithm was conducted, to improve the intelligent level of hoisting construction and realize real-time interaction in the hoisting process. The integrated data visualization display realized the virtual control of the hoisting process and effectively improved the management level.

Generally, the advantages of the LoRa technology, such as long distance, mass nodes, high security, low energy consumption, and flexible networking, makes it suitable for application scenarios of prefabricated building construction. This study showed that the application of multi-sensor information fusion method driven by the IoT to realize the intelligent hoisting control of prefabricated components is feasible and has a broad application prospect. However, at present, the research on intelligent hoisting construction of PC is still in the primary stage, and the research on intelligent hoisting equipment and management system is still in the primary stage. Improving the actual effect of information interaction and fusion and communication technology transmission analysis, and realizing the deep integration of IoT devices and other intelligent technologies to promote digital display of the whole life cycle of intelligent construction and operation of buildings would be the focus of further research and exploration.

## Figures and Tables

**Figure 1 sensors-21-00980-f001:**
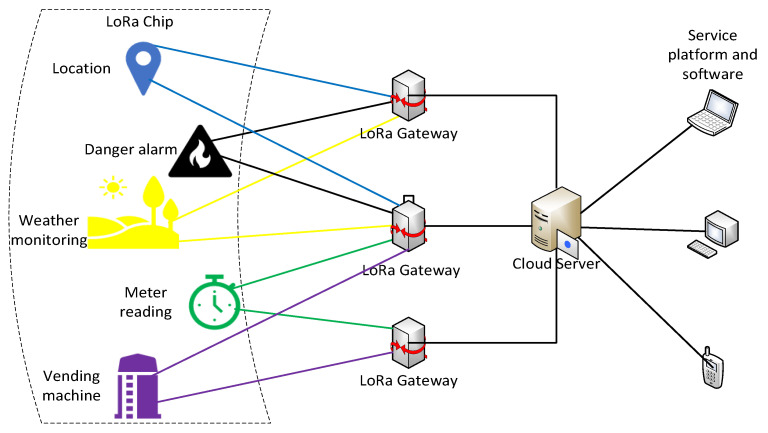
LoRaWAN network architecture.

**Figure 2 sensors-21-00980-f002:**
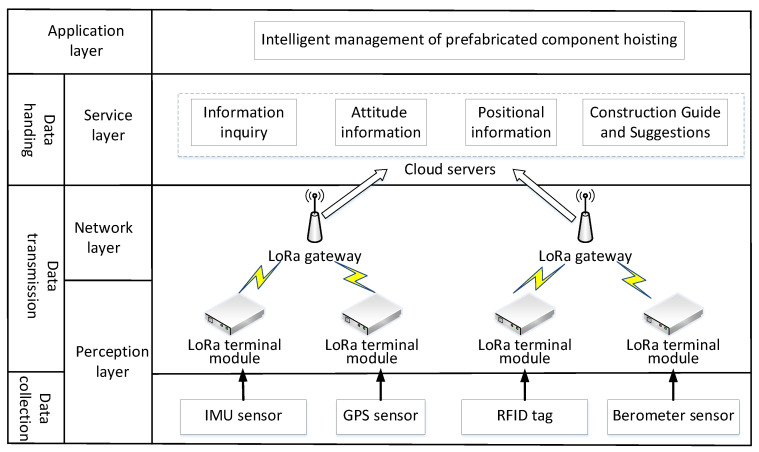
Application functional architecture.

**Figure 3 sensors-21-00980-f003:**
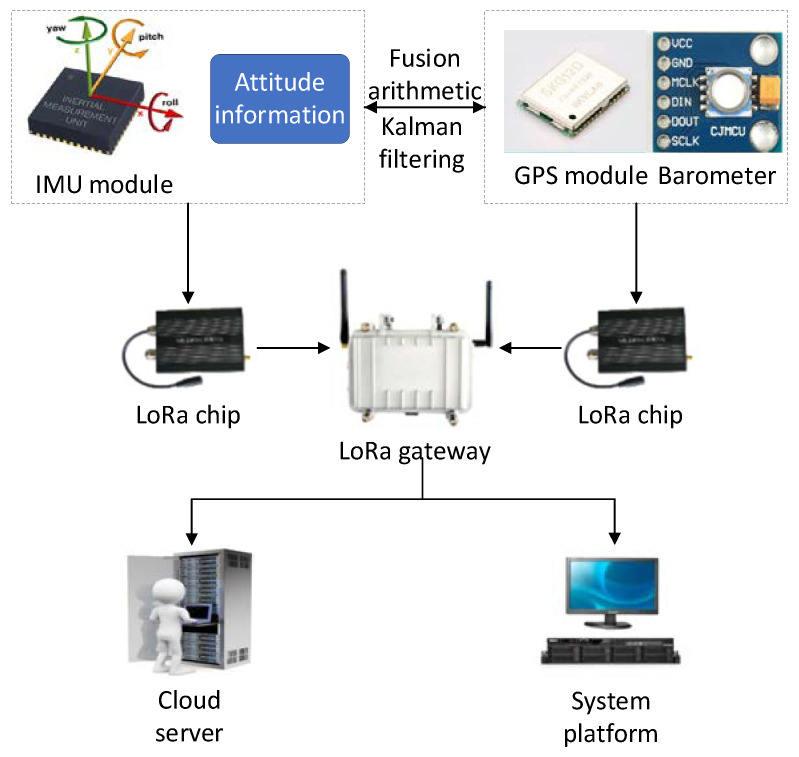
Logic diagram of the hoisting control method.

**Figure 4 sensors-21-00980-f004:**
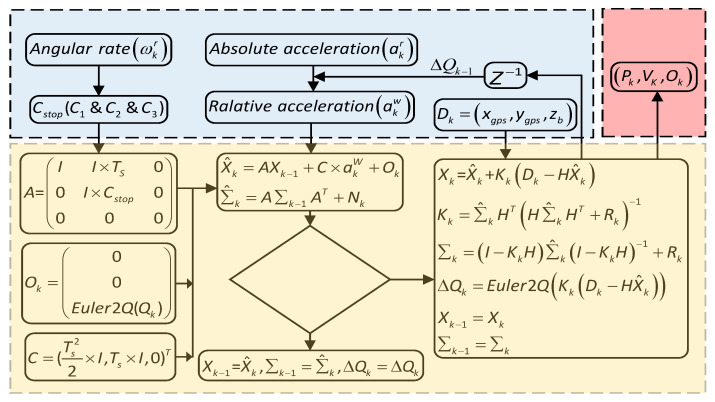
Logical diagram of the positioning algorithm.

**Figure 5 sensors-21-00980-f005:**
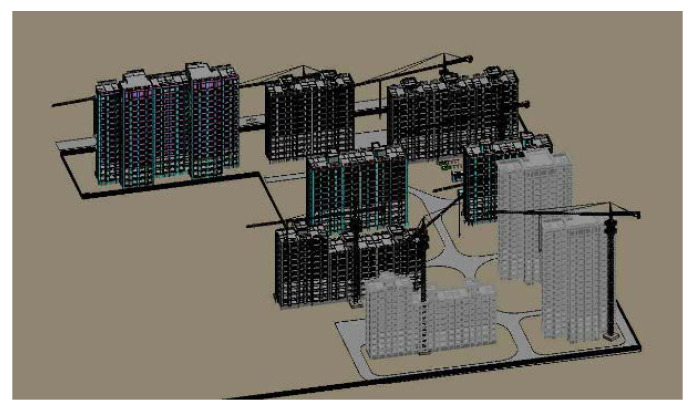
Building Information Modeling (BIM) model for field layout of construction site.

**Figure 6 sensors-21-00980-f006:**
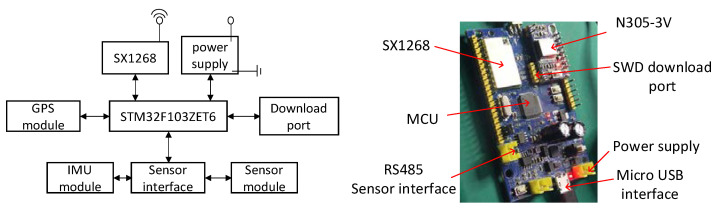
LoRa transmission terminal.

**Figure 7 sensors-21-00980-f007:**
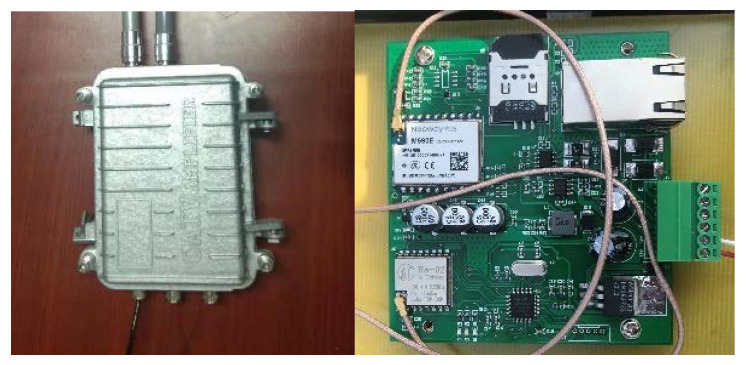
LoRa gateway installed on the construction site.

**Figure 8 sensors-21-00980-f008:**
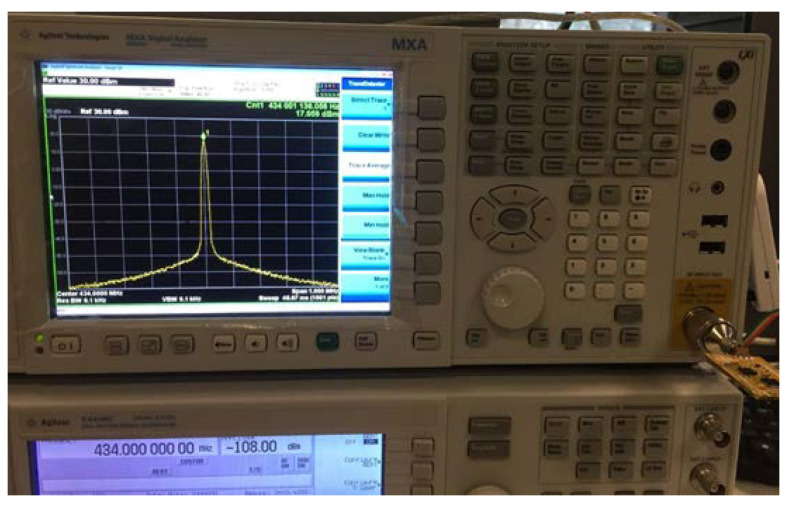
Analysis of power consumption of positioning terminal based on the LoRa network.

**Figure 9 sensors-21-00980-f009:**
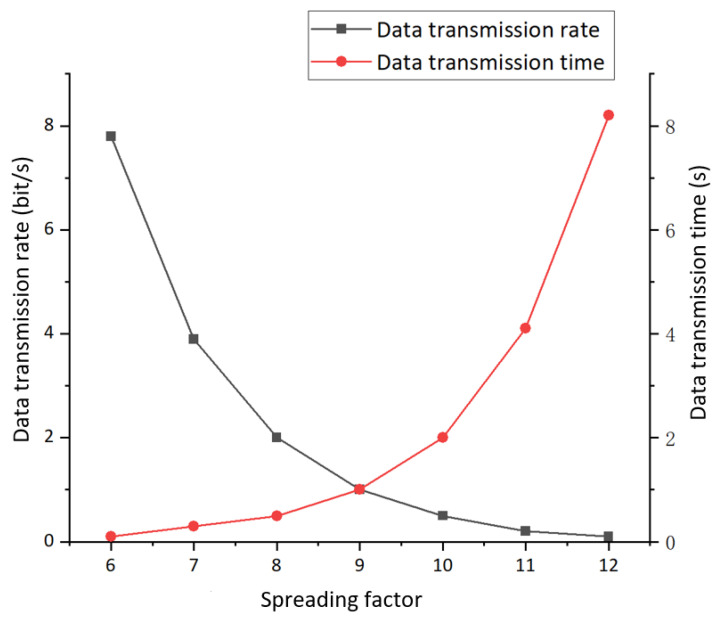
Analysis of data transmission rate and data transmission time in the LoRa network.

**Figure 10 sensors-21-00980-f010:**
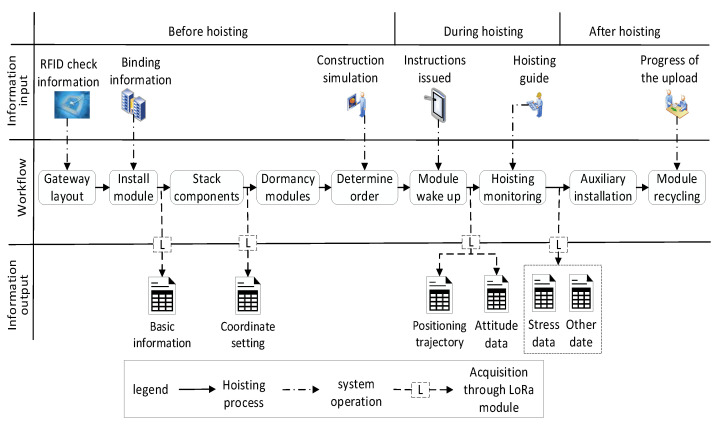
Control flow of PC hoisting.

**Figure 11 sensors-21-00980-f011:**
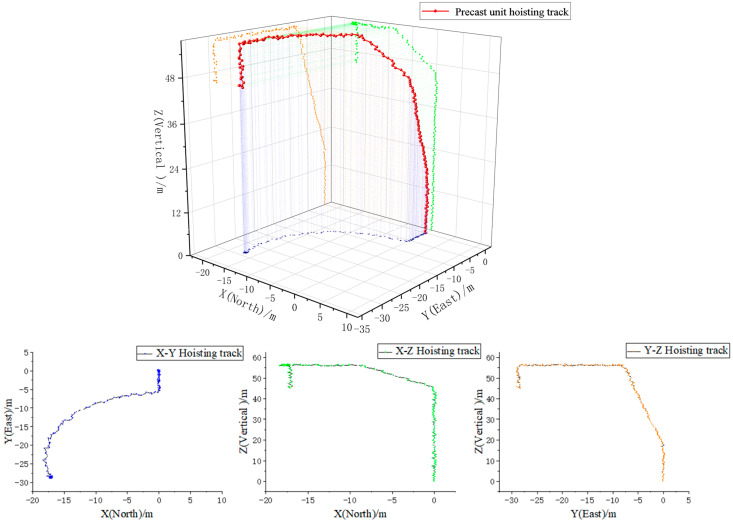
Hoisting track diagram of PC.

**Table 1 sensors-21-00980-t001:** Precast component details.

Building Number	Total Floors	Start Layer	End Layer	Assembly Layer	Single Wall Panels	Wallboards Total Number	Number of Single Floor Slab	Number of Single Stairs
10	15	4	14	11	16	176	96	4
11	11	4	10	7	4	28	50	4
12	11	4	10	7	6	42	78	6
13	11	4	10	7	4	28	52	4
14	11	4	10	7	8	56	76	4
15(1)	11	4	10	7	4	28	38	2
15(2)	8	3	7	5	8	40	76	4
16	18	4	17	14	8	112	46	2
17	18	4	17	14	8	112	46	2
18(1)	11	4	10	7	2	14	26	2
18(2)	8	3	7	5	4	20	52	4

**Table 2 sensors-21-00980-t002:** Hardware models and parameters used.

Hardware Types	Module Selection	Introduction of Module
Lora chip	Semtech SX1268	Semtech’s new long-range, low-power sub-GHz wireless transceiver chip
MCU	STM32F103ZET6	ST company under a commonly used enhanced series of microcontrollers
GPS	N305-3V	Professional level dual mode navigation and
Barometer	BMP180	A high-precision, low-power digital pressure sensor for mobile phones, GPS navigation devices.
IMU	JY901B	Intelligent attitude information measurement module developed by Witter

**Table 3 sensors-21-00980-t003:** The properties of LoRaWAN.

LoRaWAN Parameter	LoRa Values
Channel Bandwidth	125 to 500 kHz
Uplink data rate	29–50 kbps
Downlink data rate	7–50 kbps
Spreading factor	27to 212
Link budget	BMP180156 dB
Doppler sensitivity	Up to 40 ppm

## Data Availability

Data sharing not applicable.

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
