# Peer review of "Intelligent Control Method of Hoisting Prefabricated Components Based on Internet-of-Things"

_sensors, 2021, doi:10.3390/s21030980_

Round 1

Reviewer 1 Report

the proposal is interesting, but it is not developed in a clear way that indicates the main contribution that was achieved with respect to other references.
It should be clearly indicated what the contribution is in each section.
The presentation only indicates an integration of technologies, which is not sufficient for it to be published.

as particular observations, the following is indicated:

Figure 1. does not indicate the four parts that make up the LoRaWan.

2.- Condition 1: C1, C2, C3what does it represent?

3.- Equation (7) the term on the left side is confusing.

4.- The state matrix is normally obtained from the plant model, in this case it is not indicated how it was obtained.

5.- Fig 10. improve x-y-z axis legends.

6.- The final results shown in Fig 10, are not discussed, it is necessary to attach a brief description of each result shown.

7 .- The section of discussion and conclusion, does not indicate in terms of the proposal that was achieved in a significant manner with respect to the proposals indicated in the state of the art.

Author Response

The proposal is interesting, but it is not developed in a clear way that indicates the main contribution that was achieved with respect to other references.

It should be clearly indicated what the contribution is in each section.
The presentation only indicates an integration of technologies, which is not sufficient for it to be published.

as particular observations, the following is indicated:

First of all, thank you very much for taking time out of your busy schedule to read and revise my article, thank you for your valuable advice, and you have made a comprehensive correction to the research content of my paper, which will play a very important role in improving the quality of my paper.

I have carefully studied the opinions of the reviewer, and according to the suggestions, carefully revised the paper one by one as follows:

We improved the abstract, Introduction and literature review part. The integration of Lora, IMU, GPS integrated control is added in the abstract, that is, “Analysis of the existing outdated prefabricated component hoisting management methods, Lora, IMU, GPS and other sensors are used for data acquisition and transmission, the prefabricated component hoisting multi-level management and control, real intelligent control.” Section 2.3 is added to the literature review, which emphasizes the lack of research in the field of hoisting and points out the research focus of this paper, that is “Although there are many applications of IoT in the field of prefabricated building construction, few studies have been performed on the application of IoT technology in the field of prefabricated component hoisting. Moreover, because of the backwardness of the communication in the operation of the prefabricated components, the communication on the swing and position of the prefabricated components in the process of hoisting is still in the form of megaphone and walkie-talkie. Additionally, the performance of data acquisition and transmission on the construction site is backward, be-cause more advanced transmission technology is not used in the field information acquisition and transmission, thus, there is no reliable information collection and trans-mission method in the hoisting process of prefabricated components, and the visual control of prefabricated components in the construction process is obviously insufficient. Reliable information collection and transmission methods are lacking in the field of prefabricated components hoisting, which causes the obvious lack of visual control of prefabricated components during construction. To solve these problems, in the present study LoRa, IMU, GPS and other IoT related technologies is used to establish the hoisting control method of prefabricated components. The method was employed to conduct meaningful investigations as follows: (1) Hoisting process data collected according to the IoT, and realize real-time transmission of different types of data. (2) Based on the analysis of the comprehensive hoisting data, a hoisting construction management method of prefabricated components was proposed to guide the site construction personnel to conduct the hoisting construction. (3) The fusion positioning algorithm of IMU and GPS based on Kalman filter has been verified in the feasibility of lifting positioning. (4) The site management and control of the prefabricated component hoisting is realized to help the hoisting operators and managers quickly understand the status of the prefabricated component during hoisting. This innovative method, which integrates technologies such as LoRa, IMU, GPS, integrated positioning and others, improves the management level of prefabricated component hoisting, and is of great significance for future research of prefabricated building construction.”

Point 1:  Figure 1. does not indicate the four parts that make up the LoRaWan.

Response 1: And the title of the article has been modified. “3. control method” was changed to “3. Proposed Method” “3.1 LoRa technology application scheme” was changed to “LoRa WAN architecture for hoisting”. Modifications to Figure 1 indicate the four sections of LoRaWAN. Explanations for C1, C2 and C3 have been added in the paper, that is, “Through the above three conditions, the logical value of zero speed detection Cstop= C1 & C2 & C3 , where C1 , C2 , C3 are represented in Condition 1, 2, 3.” See line 264 of the paper for details.

Point 2: Condition 1: C1, C2, C3what does it represent?

Response 2: The three conditions C1, C2, C3 are used to determine whether the acceleration and angular velocity of an object are generated by disturbance, which limits the slight disturbance and excludes its influence. According to your opinions, the formula of formula 7-11 should be expressed and modified in order. See Equation (6), (7), (8), (9), (10) (11) for details.

Point 3: Equation (7) the term on the left side is confusing.

Response 3: We have modified the original formula 7.See Formula 8 for details.

Point 4: The state matrix is normally obtained from the plant model, in this case it is not indicated how it was obtained.

Response 4: There is an ambiguity in the state matrix representation according to your question. Therefore, this paper is modified from “The state matrix at time K is defined as  ” to “Using state space representation, the system’s state vector can be given as ”. In this paper, the state vector is defined by the combination of position vector, velocity vector and Angle vector.

Point 5: Fig 10. improve x-y-z axis legends.

Response 5: The x-y-z axis legend has been improved.

Point 6: The final results shown in Fig 10, are not discussed, it is necessary to attach a brief description of each result shown.

Response 6: "It can be obtained through multi-sensor information fusion and algorithm processing, as shown in Figure 11. The path of the PC presents advanced path motion in the projection of X-Y plane, and then tangential motion was conducted, and the corresponding tangential motion was adjusted in the process, and the deflection was conducted to the inside of the tangential motion. From the projection of X-Z plane and Y-Z plane, it can be seen that the vertical movement of the PC was accompanied by alternating tangential movement and radial movement, gradually shifting to the installation position. Meanwhile, it can be seen from the three-dimensional trajectory diagram of X-Y-Z that the PC presents a complex trajectory with three kinds of motions continuously combined and adjusted alternately in the traveling path. The results presented in the figure are basically consistent with the pre-hoisting planning and the usual hoisting trajectory, which reflects the timely visual display and intelligent control effect of the multi-sensor fusion processing and transmission method on the response of the PC in the hoisting process." was added to the paper to describe and analyze the results in original Figure 10.

Point 7: The section of discussion and conclusion, does not indicate in terms of the proposal that was achieved in a significant manner with respect to the proposals indicated in the state of the art.

Response 7: The discussion and conclusion of the fifth chapter are greatly improved, this is “The rapid development of new generation technologies such as the IoT and artificial intelligence has promoted the construction industry to upgrade to digital and intelligent construction. IoT technology is the key technology to realize the transformation and upgrading of prefabricated building construction as well as efficient method for the management and control of PC hoisting system. This study analyses the construction hoisting status and the application of information technology, investigates the fusion method of LPWAN and multi-sensor information integration, and introduces the key role of IoT technology in realizing intelligent control of PC hoisting, and analyses the realization effect of multi-information fusion processing. This method has the following advantages:

(1) Aiming at the application method of IoT technology in the construction stage, the integration mechanism of LPWAN and multi-sensor was explored. The real-time data feedback provides a new idea for the realization of "real-time perception, intelligent analysis and intelligent decision" in PC construction.

(2) To apply the LoRa technology and multi-sensor fusion mechanism to hoisting of PC, we first perform the deep fusion of LoRa technology and multi-sensor to realize real-time perception and upload of information, which lay a foundation for the deep visualization, real-time feedback, intelligent control and accurate execution of prefabricated building construction. Meanwhile, the ability of depth information fusion to improve the optimization control of PC during hoisting process is discussed.

(3) Improving the intelligentization of PC hoisting construction is a system engineering of multidisciplinary integration. This study takes the intelligent control of PC hoisting based on the multi-information fusion transmission of the IoT as the core, realizes the multi-information fusion transmission and processing mechanism in the hoisting process, and continuously improves the ability of intelligent control of PC hoisting.

(4) Aiming at the problem of intelligent hoisting control of PC, an integrated method driven by intelligent sensing technology and intelligent algorithm is conducted to improve the intelligent level of hoisting construction and realize real-time interaction in the hoisting process. The integrated data visualization display realizes the virtual control of hoisting process and improves the management level effectively.

Generally, the advantages of LoRa technology, such as long distance, mass nodes, high security, low energy consumption and flexible networking, make it suitable for application scenarios of prefabricated building construction. This study shows that the application of multi-sensor information fusion method driven by the IoT to realize the intelligent hoisting control of prefabricated components is feasible and has a broad ap-plication prospect. However, at present, the research on intelligent hoisting construction of PC is still in the primary stage, and the research on intelligent hoisting equipment and management system is still in the primary stage. By improving the actual effect of information interaction and fusion and communication technology transmission analysis, realizing the deep integration of IoT devices and other intelligent technologies to promote the digital display of the whole life cycle of intelligent construction and operation of buildings will be the focus of further research and exploration.”.

Reviewer 2 Report

The paper titled "Intelligent Control Method of Hoisting Prefabricated Components Based on Internet-of-Things" seems to be a quite interesting topic. Nevertheless, the way the article is present in the weakest point. Therefore this article cannot be accepted for publication. It has not been showing exactly how the proposed solution is better than the other. There is no straight comparison between the proposed solution and the already known from the literature topic.

Authors are using a lot of formulas. However, they are not used in any chart and table. So, the question is: are they needed at all in this paper? If yes, why they are not used in the next part of the paper. And if they are used, why are they not emphasis enough?

There are a lot of other flaws. Several of them are as follows:
1) References are prepared messily. For example, sometimes there is a lack of commas, authors' names are given in full or as abbreviations, publication's year is given in braces or not, etc.
2) Reference list is not prepared in the order of citing them in the text.
3) The proper acronym for the Internet of things is IoT, not IOT. It is sometimes wrongly used in the literature; however, this should not justify authors to use the wrong form. Moreover, authors are using both ways of writing it (for example, see lines 90 and 92).
4) All acronyms should be defined in the first place of use. The abstract could have acronyms' definitions; however, these definitions should be placed for sure in the right text like the Introduction and other sections.
5) Line 58 - author's surname from a small letter?
6) The last paragraph of section 1 should constitute information for the rest part of this paper.
7) After section 1, there is section 2.2, which has subsection 2.1... And where is section 2?
8) ZigBee or Zigbee? Is it the same? If yes, why is there a different way of writing it?
9) Title of section 3 is started with a small letter.
10) All formulas are given in different font types, in many cases, a lower font than the rest of the paper's text. It looks very unprofessional and sometimes strange.
11) All variables should be given in italic. Regular text is not a good idea because sometimes it gives strange sentences like, for example, "...and I represents the identity matrix" (line 221).
12) Which formula? Only one formula? (lines 221, 228). I suggest using the "equation" word instead of the "formula".
13) All figures should be placed after the first reference in the text.
14) Figure 4 should be bigger - right now, the used font is much smaller than the article's text. In fact, Figure 4 is an algorithm, and it should be more readable.
15) "kalman filter" or "Kalman filter"?
16) Table 1 is cited earlier than Figure 5. However, in the text, it is placed just after Figure 5. It is the wrong manner.
17) "Figure.6" and "Figure.7" is a wrong caption name, it should be "Figure 6" and "Figure 7".
18) All figures should be placed in the order of referring them in the right article's text.
19) Table 2 should not be split into two pages.
20) Sometimes a new sentence is started from a small letter, sometimes a big letter is used for a word in the middle of the sentence, etc.

Author Response

The paper titled "Intelligent Control Method of Hoisting Prefabricated Components Based on Internet-of-Things" seems to be a quite interesting topic. Nevertheless, the way the article is present in the weakest point. Therefore this article cannot be accepted for publication. It has not been showing exactly how the proposed solution is better than the other. There is no straight comparison between the proposed solution and the already known from the literature topic.

Authors are using a lot of formulas. However, they are not used in any chart and table. So, the question is: are they needed at all in this paper? If yes, why they are not used in the next part of the paper. And if they are used, why are they not emphasis enough?

Thank you very much for taking time out of your busy schedule to read and modify my article, and thank you very much for your valuable suggestions. We have studied your suggestion very carefully and made detailed and comprehensive revision and improvement to the paper. Your suggestion plays a very important role in improving the quality of the paper. Thanks again.

We have improved abstracts, introductions, literature reviews, and discussions and conclusions, and highlighted the research adopts IoT are the lack of technology to strengthen the prefabricated construction hoisting field and has a broad application prospect. The introduction has been comprehensively revised and enhanced to highlight the relevance of this study to other studies. In addition, "2.3 Research emphasis and novelty" was added in the literature review to emphasize the contribution and importance of this Research. This study proposed the multiple sensor fusion method based on Internet of things improve the prefabricated construction hoisting management informationization, visualization and intelligent level, to promote prefabricated building digital, intelligent construction is of great significance.

Point 1: References are prepared messily. For example, sometimes there is a lack of commas, authors' names are given in full or as abbreviations, publication's year is given in braces or not, etc.

Response 1: References have been reorganized, revised and checked as required.

Point 2: Reference list is not prepared in the order of citing them in the text.

Response 2: The citation of references has been reordered.

Point 3: The proper acronym for the Internet of things is IoT, not IOT. It is sometimes wrongly used in the literature; however, this should not justify authors to use the wrong form. Moreover, authors are using both ways of writing it (for example, see lines 90 and 92).

Response 3: The article IOT has been uniformly modified to IoT. At the same time, this kind of problem is carefully examined.

Point 4: All acronyms should be defined in the first place of use. The abstract could have acronyms' definitions; however, these definitions should be placed for sure in the right text like the Introduction and other sections.

Response 4: The initials have been used in accordance with the requirements.

Point 5: Line 58 - author's surname from a small letter?

Response 5: It has been changed to uppercase.

Point 6: The last paragraph of section 1 should constitute information for the rest part of this paper.

Response 6: A description of the rest of the article has been appended to the end of section 1, that is “The remainder of the paper is organized as follows: Section 2 presents a literature review about the previous works in related area. Section 3 describes in detail the proposed multi-sensor sensing hoisting control method. Section 4 presents a sensor test and a field test of the proposed method applied to a real project. Section 5 discusses the results and concludes the paper.”.

Point 7: After section 1, there is section 2.2, which has subsection 2.1... And where is section 2?

Response 7: An error occurred in heading 2 and has been corrected.

Point 8: ZigBee or Zigbee? Is it the same? If yes, why is there a different way of writing it?

Response 8: It's the same. It's been modified.

Point 9: Title of section 3 is started with a small letter.

Response 9: The title of section III has been revised.

Point 10: All formulas are given in different font types, in many cases, a lower font than the rest of the paper's text. It looks very unprofessional and sometimes strange.

Response 10: We modified all the formulas and variables to adopt the MathTpye typeface of "Palatino Linotype, italic."

Point 11: All variables should be given in italic. Regular text is not a good idea because sometimes it gives strange sentences like, for example, "...and I represents the identity matrix" (line 221).

Response 11: All scalars have been changed to italics.

Point 12: Which formula? Only one formula? (lines 221, 228). I suggest using the "equation" word instead of the "formula".

Response 12: “formula” has been modified to “equation”.

Point 13: All figures should be placed after the first reference (sentence) in the text.

Response 13: We have adjusted the position of all charts accordingly.

Point 14: Figure 4 should be bigger - right now, the used font is much smaller than the article's text. In fact, Figure 4 is an algorithm, and it should be more readable.

Response 14: Figure 4 has been redrawn and improved, with larger fonts and aesthetics.

Point 15: "kalman filter" or "Kalman filter"?

Response 15: Has been uniformly modified to “Kalman filter”.

Point 16: Table 1 is cited earlier than Figure 5. However, in the text, it is placed just after Figure 5. It is the wrong manner.

Response 16: The order of Table 1 and Figure 5 has been adjusted.

Point 17: "Figure.6" and "Figure.7" is a wrong caption name, it should be "Figure 6" and "Figure 7".

Response 17: “Figure .6 and Figure .7” have been modified to “Figure 6 and Figure 7”.

Point 18: All figures should be placed in the order of referring them in the right article's text.

Response 18: All the charts have been rearranged in order of reference.

Point 19: Table 2 should not be split into two pages.

Response 19: The position of Table 2 has been readjusted.

Point 20: Sometimes a new sentence is started from a small letter, sometimes a big letter is used for a word in the middle of the sentence, etc.

Response 20: The case of the sentence in the article has been rechecked.

Reviewer 3 Report

The novelty and the contribution of the paper could be highlighted more clearly. The authors should describe the purpose and contribution of their research in the introductory section.

One third of the references cited in the state-of-the-art section are more than 5-10 years old. One can therefore say that the study is no longer up to date. The reference list should be updated with more recent scientific papers in journals, conferences, or books.

A large part of the paper, presents different known concepts, an introduction, and a literature review: Inertial measurement unit; Attitude (Altitude?) control method; Position monitoring method.

The "Hardware selection and development" chapter presents most of all only images of the equipment and modules instead of including charts, communication protocol description, hardware and software architecture. This chapter must be completely rewritten before the contribution can be considered for publication. Figure 7 (a, b and c) can be removed.

Several included figures have a poor quality. Please consider to improve the quality of the figures: 2, 3, 4, 5, 6, 8 and 10.

The addition of an estimation of the power consumed by the complete system (measurement, acquisition, processing, and transmission through LoRa) and of the cost could be of interest for the readers.

I recommend the authors to insist more on the testing scenarios and on the analysis and interpretation of the obtained results rather than on previously known information.

It is not clear what novelty the paper brings compared to other implementations. If we talk about low cost, the authors do not demonstrate through a comparative table that the cost of the proposed system is lower than to the one of other existing systems.

Author Response

Thank you very much for taking the time to read and modify my article, and thank you for your valuable suggestions. You have made a comprehensive revision of the research content of my paper, which will serve to improve the quality of the paper. We have made a number of changes in accordance with your comments, as shown below.

Point 1: The novelty and the contribution of the paper could be highlighted more clearly. The authors should describe the purpose and contribution of their research in the introductory section.

Response 1: The introduction has been greatly modified to highlight the purpose and contribution of the research. Meanwhile, "2.3 Research emphasis and novelty" was added in the literature review to emphasize the research focus and major contribution in this paper.

Point 2: One third of the references cited in the state-of-the-art section are more than 5-10 years old. One can therefore say that the study is no longer up to date. The reference list should be updated with more recent scientific papers in journals, conferences, or books.

Response 2: The cited literature has been modified and replaced, and the literature that was published too long has been removed. The deleted literature is as follows: (3) Ghang Lee, Joonbeom Cho and Sungil Ham, et al, “A BIM- and sensor-based tower crane navigation system for blind lifts,” Automation in Construction, Vol.26(2012), pp.1-10. (9) Fen Wang, Fengyou He, “Study of Hoist Perception System Based on IOT Technology”.// WISM. Proceedings - 2010 International Conference on Web Information Systems and Mining. Hainan, China: IEEE,2010:357-360. (18) William H. Baird. “An introduction to inertial navigation,” American Journal of Physics. Vol.77, No9(2009), pp.844-847. (20) Tam V W Y , Fung I W H, “Tower crane safety in the construction industry: A Hong Kong study,” Safety Science, Vol.49, No.2(2011),pp. 208-215. (30) Kong X, “INS algorithm using quaternon model for low cost IMU,” Robotics and Autonomous Systems, Vol.46, No.4(2004), pp.221-246. (35) He X , Chen Y and Liu J, “Development of a low-cost integrated GPS/IMU system,” IEEE Aerospace & Electronic Systems Magazine, Vol.13, No.12(1998), pp.7-10. (36) D. Guo, L. Wu and J. Wang, et al, “Use the GPS/IMU New Technology for Photogrammetric Application,” IEEE International Conference on Geoscience & Remote Sensing Symposium, IEEE, 2006. Meanwhile, the introduction has been greatly revised and the latest research literature in a number of related fields has been added. For example, references [1-8], [10-13], etc.

Point 3: A large part of the paper, presents different known concepts, an introduction, and a literature review: Inertial measurement unit; Attitude (Altitude?) control method; Position monitoring method.

Response 3: We corrected the relevant concepts uniformly, and revised the titles 3.2 and 3.3 to "IMU information collection" and "Sensors information fusion processing" respectively.

Point 4: The "Hardware selection and development" chapter presents most of all only images of the equipment and modules instead of including charts, communication protocol description, hardware and software architecture. This chapter must be completely rewritten before the contribution can be considered for publication. Figure 7 (a, b and c) can be removed.

Response 4: We rewrote the chapter of "Hardware Selection and Development". On the basis of introducing the image types of devices and modules, we added the content of terminal hardware architecture, communication protocol description, important communication parameters, hardware power consumption test. And we remove the original figure 7.

Point 5: Several included figures have a poor quality. Please consider to improve the quality of the figures: 2, 3, 4, 5, 6, 8 and 10.

Response 5: We made adjustments and improvements to the original figures 2,3,4,5,6,8,10, among which the original figures 2,4 was redrawn.

Point 6: The addition of an estimation of the power consumed by the complete system (measurement, acquisition, processing, and transmission through LoRa) and of the cost could be of interest for the readers.

I recommend the authors to insist more on the testing scenarios and on the analysis and interpretation of the obtained results rather than on previously known information.

Response 6: Your insights are excellent. Thank you very much for your advice. LoRa technology has the advantages of low power consumption and low cost. In "4.2. Hardware Selection and Development", this paper introduces the power test of the developed Hardware, and verifies the characteristics of low power consumption of LoRa technology. This study takes assembly construction and hoisting as an example to study the intelligent control of LoRa technology and multi-sensor fusion. The idea and feasibility of the application of LoRa technology in prefabricated buildings are verified. Based on the advantages of low cost and low power consumption of LoRa technology, we plan to compare and analyze the power consumption and cost of application of LoRa technology with that of traditional information transmission network in future research. Meanwhile, in "4.3. Analysis of Hoisting Process", the effect Analysis of multi-sensor application based on LoRa technology is added to verify the feasibility of LoRa technology and this method in assembly construction management.

Point 7: It is not clear what novelty the paper brings compared to other implementations. If we talk about low cost, the authors do not demonstrate through a comparative table that the cost of the proposed system is lower than to the one of other existing systems.

Response 7:The introduction has been greatly modified to highlight the purpose and contribution of the research,Meanwhile, a section on "2.3 Research emphasis and adventures" is added to highlight the Research focus of this paper. The "5.Discussion and Conclusion" has been modified and improved, and the main conclusions and contributions of this paper have been explained. This study aims to improve the integration of the new technology of IoT and multi-sensor to improve the intelligent level of prefabricated building construction management. Through field verification, this method has a certain value and provides a new idea and method for the new technology of IoT in the field of assembly construction management.

Reviewer 4 Report

In this article, the authors present a practical operation of LORA to develop an Intelligent Control Method of Hoisting Prefabricated Components.

The practical use and realistic experiments are the strong aspects of the paper.
However, it is not clear the architecture the authors propose. For example, are you proposing the entire architecture of figures 1 and 2?

Given the current text, one is not able to reproduce the paper or the LORA system authors propose. You must show how parts of the system integrate, exchange messages, etc.

I suggest authors to clear delimit their contribution. Tell readers clearly what they are getting from the literature and what is new in fact. In some parts of the text, the contributions of the paper are overestimated.

The paper text itself must be carefully revised. There are pieces of text from a previous article template.

Author Response

Thank you very much for taking time out of your busy schedule to read and modify my article, and thank you very much for your valuable suggestions. We have carefully studied your suggestions and have made detailed and comprehensive revisions and improvements to the paper. Your suggestion plays a very important role in improving the quality of the paper. The description of the improvement we have made in response to your suggestion as follow:

Point 1: In this article, the authors present a practical operation of LORA to develop an Intelligent Control Method of Hoisting Prefabricated Components. The practical use and realistic experiments are the strong aspects of the paper. However, it is not clear the architecture the authors propose. For example, are you proposing the entire architecture of figures 1 and 2?

Response 1:We modified Figure 1 and Figure 2, and made a lot of modifications to "4.2. Hardware Selection and Development", adding content such as Hardware architecture test, information transmission, communication parameters, etc., to explain how the information transmission in Figure 1 and Figure 2 is realized in the case.

Point 2: Given the current text, one is not able to reproduce the paper or the LORA system authors propose. You must show how parts of the system integrate, exchange messages, etc.

Response 2: In this paper, we have added about the system hardware design and selection, hardware testing, data transmission and so on. Meanwhile, the results of multi-sensor fusion are further analyzed.

Point 3: I suggest authors to clear delimit their contribution. Tell readers clearly what they are getting from the literature and what is new in fact. In some parts of the text, the contributions of the paper are overestimated.

Response 3: We have revised the title and content of introduction, literature review, hoisting case, discussion and conclusion. The introduction explains the relationship between this study and other relevant literatures. In the literature review section, "2.3 Research emphasis and novelty" is added to highlight the major contributions of the research. In the lifting case part, the hardware and LORA network transmission are enriched and the information processing results are analyzed. In the section of discussion and conclusion, it is rewritten to explain the research conclusions and major contributions made by this study.

Point 4: The paper text itself must be carefully revised. There are pieces of text from a previous article template.

Response 4: We have carefully modified the paper text itself and removed the redundant template text. At the same time, the formula font, font capitalization, abbreviation, the format of reference and other contents in the article and comprehensive and detailed modification.

Round 2

Reviewer 1 Report

The authors present a version of the article, where the observations indicated in the revision phase were taken into account, it is considered that the improvements allow having a new adequate proposal for its possible publication.

Author Response

Point 1: The authors present a version of the article, where the observations indicated in the revision phase were taken into account, it is considered that the improvements allow having a new adequate proposal for its possible publication.

Response 1: Thank you very much for your suggestions on the revision of this manuscript. We have revised and improved it according to your opinions. Your suggestions have played an important role in improving the quality of this manuscript. Thanks again from the bottom of my heart.

Reviewer 2 Report

The article "Intelligent Control Method of Hoisting Prefabricated Components Based on Internet-of-Things" has been significantly improved, and right now, it can be considered for publication. The authors referred to all my guidelines and tried to address all my comments. However, there are still some flaws that can improve the paper. They are as follows:

  1. Definition of the RFID acronym was in the first version of the article. Now in the second version of this article, the definition of this acronym is missing.
  2. Other acronyms are used without definition, for example, BIM, UWB.
  3. Title of subsection 3.3.1 should start on the new page.
  4. In equations where are listed cases (for example, (7), (9), and so on), there is no need to add a closing buckle. The left buckle is enough.
  5. In line 279, there should be no indent because it is a continuation of the previous line (in fact, it is an explanation of variables used in equation (26)).
  6. Figure 4 should be placed in the text just after the paragraph given in lines 283-284.
  7. In line 295, please add a hard-space. It allows putting "Figure 5" in one line. Right now, only "5)." in line 296 looks a little bit strange.
  8. In Table 1, there should be "Total floors" (the plural form).
  9. In Table 2, please add horizontal lines separating each row, similar to those in Table 1. It allows distinguishing information included in the last column.
  10. Figure 10 is still too wide, and it falls into the margin. Maybe rotating it 90 degrees allows to put it into the text area. Then authors can enlarge the font a little bit to make it more readable?
  11. Line 485 - page "1118-"? It should be without "-" at the end.
  12. Line 506 - as in a previous suggestion.

Author Response

Point 1: The article "Intelligent Control Method of Hoisting Prefabricated Components Based on Internet-of-Things" has been significantly improved, and right now, it can be considered for publication. The authors referred to all my guidelines and tried to address all my comments.

Response 1: Thank you very much for your suggestions on the revision of this manuscript. Your suggestion for revision has played an important role in improving the quality of this manuscript. Heartfelt thanks again for your help with this article.

Point 2: Definition of the RFID acronym was in the first version of the article. Now in the second version of this article, the definition of this acronym is missing.

Other acronyms are used without definition, for example, BIM, UWB.

Response 2: We have added the definitions of the acronyms RFID, BIM, GPS, UWB, GPS, NB-IoT, and LoRa.

Point 3: Title of subsection 3.3.1 should start on the new page.

Response 3: Heading 3.3.1 has been repositioned to start from a new page.

Point 4: In equations where are listed cases (for example, (7), (9), and so on), there is no need to add a closing buckle. The left buckle is enough.

Response 4: Formula 7, 9, 11, 12, 14, 15, 18, 20, 21, 22, 23, 24, 26 have been changed to the left buckle and their positions have been adjusted.

Point 5: In line 279, there should be no indent because it is a continuation of the previous line (in fact, it is an explanation of variables used in equation (26)).

Response 5: We have revised to remove indent.

Point 6 Figure 4 should be placed in the text just after the paragraph given in lines 283-284.

Response 6: We have modified the position of Figure 4.

Point 6: In line 295, please add a hard-space. It allows putting "Figure 5" in one line. Right now, only "5)." in line 296 looks a little bit strange.

Response 6: We have revised it as required.

Point 7: In Table 1, there should be "Total floors" (the plural form).

Response 7: It has been modified to plural.

Point 8: In Table 2, please add horizontal lines separating each row, similar to those in Table 1. It allows distinguishing information included in the last column.

Response 8: Horizontal rules have been added to Tables 2 and 3 as required.

Point 9: Figure 10 is still too wide, and it falls into the margin. Maybe rotating it 90 degrees allows to put it into the text area. Then authors can enlarge the font a little bit to make it more readable?

Response 9: The font in Figure 10 has been enlarged and adjusted to fit into a text area.

Point 10: Line 485 - page "1118-"? It should be without "-" at the end. Line 506 - as in a previous suggestion.

Response 10: These two places have been modified as "1-17" and "1-15" respectively.

Reviewer 3 Report

I followed the review questions and the way in which the authors responded and made the necessary improvements to the content of the paper. The authors have clarified all the aspects and they have modified the paper accordingly.

Author Response

Point 1: I followed the review questions and the way in which the authors responded and made the necessary improvements to the content of the paper. The authors have clarified all the aspects and they have modified the paper accordingly.

Response 1: Thank you very much for your suggestions on the revision of this manuscript. We have revised and improved this manuscript according to your opinions. Your suggestions have played an important role in improving the quality of this manuscript. Heartfelt thanks again for your help with this manuscript.

Reviewer 4 Report

In this version, the authors addressed all my previous questions.
In special, they have modified the introduction section and the architecture figures to delimit their contribution.
The result sections also bring new experiments, with good insights (e.g., the tradeoff they present in figure 9).
In my opinion, the paper is ready to be accepted.

Author Response

Point 1: In this version, the authors addressed all my previous questions.

In special, they have modified the introduction section and the architecture figures to delimit their contribution.

The result sections also bring new experiments, with good insights (e.g., the tradeoff they present in figure 9).

In my opinion, the paper is ready to be accepted.

Response 1: Thank you for your suggestions on the revision of this manuscript and your affirmation of our work. Your suggestions have played an important role in improving the quality of this manuscript. Heartfelt thanks again for your help with this manuscript.